

# Sensitivity study of the Regional Climate Model RegCM4 to
# different convective schemes over West Africa
Brahima KONÉ[1], Arona DIEDHIOU[1, 2], N'datchoh Evelyne TOURÉ[1], Mouhamadou Bamba
SYLLA[3], Filippo GIORGI [4], Sandrine ANQUETIN[2], Adama BAMBA[1], Adama DIAWARA[1],
Arsene Toka KOBEA[1]
[1]LAPAMF, Université Félix Houphouët Boigny, Abidjan, Côte d'Ivoire
[2]Univ. Grenoble Alpes, IRD, CNRS, Grenoble INP, IGE, F-38000 Grenoble, France
[3]WASCAL Centre of Competence, Ouagadougou, Burkina Faso
[4]International Centre for Theoretical Physics (ICTP), Trieste, Italy
*Correspondence to:* Arona DIEDHIOU (arona.diedhiou@ird.fr)
**Abstract.** The latest version of RegCM4 with CLM4.5 as land surface scheme was used to
assess the performance and the sensitivity of the simulated West African climate system to
different convection schemes. The sensitivity studies were performed over the West Africa
domain from November 2002 to December 2004, at spatial resolution of 50km x 50km and
involved five (5) convective schemes: (i) Emanuel; (ii) Grell; (iii) Emanuel over land and Grell
over ocean (Mix1); (iv) Grell over land and Emanuel over ocean (Mix2); and (v) Tiedtke. All
simulations were forced with ERA-Interim data. Validation of surface temperature at 2m and
precipitation were conducted using respectively data from the Climate Research Unit (CRU)
and Global Precipitation Climatology Project (GPCP) during June to September (rainy season).
Quantitative assessment of the sensitivity tests were carried out using the mean bias, the pattern
correlation coefficient, the root mean square difference, the probability density function of the
temperature bias and the Taylor diagram. Results revealed a better performance of the
configuration with Emanuel convection scheme to simulate the spatial and temporal variability
of the temperature and the precipitation. Therefore, the configuration of RegCM4 with CLM4.5
as land surface model and implementing Emanuel convective scheme is recommended for the
study of the West African climate system.



## 1 Introduction

Agriculture over West Africa relies mainly on rainfall and is strongly dependent on the West
African monsoon. Therefore, the onset, cessation and the amount of expected precipitation
associated with the West African Monsoon are of great importance for farmers and accurate
simulation and prediction of rainfall and temperature are crucial for various sectors, such as
agriculture, energy and health, and for decision-makers. Rainfall over West Africa is strongly
related to the meridional migration of the Inter-Tropical zone of convergence (ITCZ) and is
modulated by successive active and inactive phases of the monsoon system (Sultan et al., 2003a;
Janicot et al., 2011). After a quasi-stationary position around 5° N between mid-April and end
of June, the rainfall maxima present an abrupt shift toward the north to hold another quasi-
stationary position around 11°N in July-August, bringing precipitation over Central Sahel
region (Sultan and Janicot, 2000). This abrupt northward shift is the monsoon ''onset'' over the
Sahel and contrasts with the smooth southward retreat of the ITCZ, followed by the second
rainy season over the Guinean Coast in October–November (Sultan et al., 2003b; Janicot et al.,
2011). In addition, atmospheric circulations through African Easterly Jet (AEJ), Tropical
Easterly Jet (TEJ) and their interaction with convection play an important role in the West
African Monsoon (WAM) system (Nicholson 2013) and modulate the summer rainfall (Sylla
et al., 2013a). Various climate modeling tools have been applied over West Africa for studying
and better understanding of the WAM.
General circulation models (GCMs) are unable to include the effects of regional features (Xue
et al., 2010) due to their relatively coarse resolution. Regional Climate Models (RCMs) are
relevant tools for this purpose since they allow land surface heterogeneity and fine-scale forcing
such as complex topography and vegetation variations (Paeth and al., 2006). Moreover,
previous studies have shown that they are able to reasonably simulate the WAM climatology
(Kamga and Buscarlet, 2006; Sylla et al., 2009) and its variability (Diallo et al., 2012). RCMs
contributed to improve our knowledge of the interactions between atmospheric and surface
factors affecting the precipitation (Sylla et al., 2011; Browne and Sylla, 2012), of the influence
of external forcing such as Sea Surface Temperature (SST, Paeth and A. Hense, 2004), dust
(Konare et al., 2008; N'Datchoh et al., 2017) and land-use changes on the dynamic of the
monsoon system (Abiodun et al., 2012; Zaroug et al., 2012).
RegCM versions (Giorgi et al., 2012; Pal et al., 2007) are the one of the most commonly used
among the large range of RCMs to study the climate of West African and of many regions of
the world. Compared with the previous version (RegCM3; Pal et al., 2007), the latest release



(RegCM4) has been improved with substantial development of the software code and of the
physical representations (Giorgi et al., 2012) and with the introduction of CLM (version 3.5
and 4.5) as an option to describe land surface processes. Previously it was Biosphere-
Atmosphere Transfer Scheme (BATS; Dickinson et al., 1993) only which was used as land
surface model. Many studies have shown that the model performs well when using BATS over
the West Africa (Sylla et al., 2009; Diallo et al., 2013) but CLM offers improvements in the
land-atmosphere exchanges of moisture and energy and in the associated surface climate
feedbacks (Steiner et al., 2009). Nonetheless it was shown over India that CLM use may lead
to a weaker performance of RegCM than BATS (Halder and al., (2015). Thus, the performance
of RegCM4 when using CLM (RegCM4-CLM4.5) needs to be assessed and sensitivities tests
have to be conducted on physical processes parameterization to find the optimal configuration
of the RCM for a given region and to give the relevant information to RCM users.
Among different physical processes in climate models, the convective parameterization is
usually considered as the most important when simulating the monsoon rainfall (Im et al., 2008;
Leung et al., 2004). Simulations of regional climate are very sensitive to physical
parameterization schemes, particularly over the tropics where convection plays a major role in
monsoon dynamics (Singh et al., 2011; Srinivas et al., 2013; Gao et al., 2016). One of the main
sources of uncertainties in climate prediction is related to the representation of the clouds, which
mainly influences the energy response of the models to a disturbance (Soden and Held, 2006;
IPCC, 2007). Thus, implementing appropriate convective scheme in dynamic models is needed
for realistic simulations.
Several sensitivity studies using previous version of RegCM have been conducted over Africa.
Meinke et al., (2007) and Djiotang and Kamga (2010) showed that in West Africa, the monsoon
precipitations are sensitive to the choice of cumulus parameterization and closure schemes.
Brown and Sylla (2012) performed a sensitivity study of RegCM3 to the domain size over West
Africa and showed that a large domain is required to capture variability of summer monsoon
rainfall and circulation features. Recent study by Adeniyi (2014) using version 4 of RegCM
indicated that all convective schemes give good spatial representation of rainfall with biases
over West Africa. Komkoua and al., (2016) found that the last release of RegCM implementing
Grell as convective scheme with Arakawa-Schubert closure assumption is more suitable to
downscale the diurnal cycle of rainfall over Central Africa. However, none of these studies have
attempted to investigate a sensitivity study of the Regional Climate Model (RegCM4) to the
convective scheme over West Africa with CLM4.5 as the land surface model.





This study investigates the performance of RegCM4-CLM4.5 over West Africa using different
convection schemes in the aim to identify the "best" configuration option for the region. The
paper is structured as follows: the description of the model, data and numerical experiments
used to investigate the RegCM4 performance are described in Section 2; Section 3 analyzes and
discusses the model's performance under different convection processes; and the main
conclusions are summarized in Section 4.

**2 Model description, observation datasets and numerical experiments**
**2.1 Model description and datasets.**
The 4th generation of the ICTP RegCM (hereafter RegCM4) is used in this study. RegCM is a
limited-area model using a terrain-following σ-pressure vertical coordinate system and an
Arakawa B-grid finite differencing algorithm (Giorgi et al., 2012). The model's dynamical
component is derived from the hydrostatic version of the Pennsylvania State University
Mesoscale Model version 5 (MM5; Grell et al., 1994) with improvements on the coupling with
an advanced and complex land surface model (CLM3.5 and CLM4.5; Oleson et al., 2008 and
2013). In the version used here, the radiation scheme is derived from the NCAR global model
CCM3 (Kiehl et al., 1996) and includes representation of aerosols following Solmon et al.,
(2006) and Zakey et al., (2006). Turbulent transports of momentum, water vapor and sensible
heat in the planetary boundary layer over land and ocean are computed as Holtslag et al., (1990),
which allows nonlocal transport in the convective boundary layer. The large-scale precipitation
scheme of Pal et al., (2000) referred as SUBgrid EXplicit moisture scheme (SUBEX) includes
the subgrid variability in clouds (Sundqvist and al., 1989) and the evaporation and accretion
processes for stable precipitation. Ocean surfaces fluxes of momentum, heat and moisture are
represented using the scheme of Zeng and al., (1998) with a drag coefficient-based bulk
aerodynamic procedure and considering the influence of surface friction velocity on roughness
length computed following Smith (1988) and Brutsaert (1982), respectively for momentum and
heat (and also moisture).
The soil-vegetation-atmosphere interaction processes are parameterized using Community
Land Model (CLM version 4.5; Oleson et al., 2013). CLM4.5 presents in each grid cell the
possibility to have fifteen soil layers, up to five snow layers, five different land unit types  and
sixteen different plant functional types (Lawrence et al., 2011; Wang et al., 2016). RegCM4-
CLM4.5 proposes five different convective schemes (Im et al., 2008); Giorgi et al., 2012): the
modified-Kuo scheme (Anthes et al., 1987), the Tiedtke scheme (Tiedtke 1989), the Emanuel
scheme (Emanuel 1991), the Grell scheme (Grell 1993) and the Kain-Fritsch scheme (Kain-



Fritsch, 1990; Kain 2004) with the possibility to combine different schemes over ocean and
land (called as 'mixed' convection).
For this sensitivity study, the model was run at its standard configuration with 18 vertical sigma
layers (model top at 50 hPa) and with initial and boundary conditions provided by the European
Centre for Medium Range Weather Forecasts reanalysis ERA-interim (Simmons et al., 2007;
Uppala et al., 2008) at an horizontal resolution of 50 km and a temporal resolution of 6 hours
(00:00, 06:00, 12:00 and 18:00 UTC). Sea-surface temperatures (SST) were from NOAA
optimal interpolation weekly SST data (Reynolds et al., 2007). The terrain characteristics
(topography and land use data) were derived from United States Geological Survey (USGS)
and Global Land Cover Characterization (GLCC; Loveland et al., 2000) respectively at 10 min
horizontal resolution.
We focus our analysis on the precipitation and on the air temperature at 2m in the summer of
June-July-August-September (JJAS) over mainland West Africa. To reduce uncertainty due to
lack of surface climate observations over the region (Nikulin et al., 2012; Sylla et al., 2013a),
the simulated precipitation is validated using two observational datasets including the monthly
mean precipitation at 2.5° horizontal resolution from Global Precipitation Climatology Project
(GPCP; Adler et al., 2003) available from 1979 to present and the 0.25° high resolution dataset
of Tropical Rainfall Measuring Mission 3B43V7 (TRMM) available from 1998 to 2013
(Huffman et al.2007). The simulated 2m temperature is validated using also two observational
datasets including the Climate Research Unit (CRU) time series version 3.20 gridded at 0.5°
horizontal resolution from the University of East Anglia and available respectively from 1901
to 2011 (Harris et al., 2013), and the University of Delaware version 3.01 (UDEL) gridded
dataset at 0.5° horizontal resolution available from 1900 to 2010 (Legates and Willmott, 1990).
The simulated atmospheric fields are compared with ERA-Interim reanalysis available from
1979 to present at 1.5° horizontal resolution (Dee et al., 2011). All products are remapped onto
the RegCM4 grid (0.44°×0.44°) using a bilinear interpolation method to facilitate the
comparison (Nikulin et al., 2012). The model's performance is further examined in four sub-
regions (Fig. 1), each with different characteristics of the annual cycle of rainfall: Central Sahel
(10°W–10°E; 10°N–16°N), West Sahel  (18°W–10°W; 10°N–16°N), Guinea Coast (15°W–
10°E; 3°N–10°N) and West Africa (20°W–20°E; 5°S–21°N).

**2.2 Convective schemes**
The convective precipitation parameterizations used in this study are Tiedke (1989), Emanuel
(1991) and Grell (1993) schemes.



The Emanuel (1991) scheme assumes that the mixing in clouds is highly episodic and
inhomogeneous (in contrary to a continuous entraining plume) and takes into account
convective fluxes based on an idealized model of sub-cloud-scale updrafts and downdrafts.
Convection is triggered when the level of neutral buoyancy is greater than the cloud base level.
Between these two levels, air is lifted and a fraction of the condensed moisture forms
precipitation while the remaining fraction forms the cloud. The cloud is supposed to mix with
the air from the environment according to a uniform spectrum of mixtures that ascend or
descend to their respective levels of neutral buoyancy. The mixing entrainment and detrainment
rates depend on the vertical gradients of buoyancy in clouds. Emanuel scheme includes a
formulation of the auto-conversion of cloud water into precipitation inside cumulus clouds.
In the Grell (1993) scheme, deep convective clouds are represented by an updraft and a
downdraft that are undiluted and mix with environmental air only in cloud base and top. Heating
and moistening profiles are derived from latent heat released or absorbed, linked with the
updraft-downdraft fluxes and compensating motion (Martinez-Castro et al., 2006). Two types
of Grell scheme convective closure assumption can be found in RegCM4. In the Arakawa–
Schubert (1974) closure (AS), a quasi-equilibrium condition is assumed between the generation
of instability by grid-scale processes and the dissipation of instability by sub-grid (convective)
processes. In the Fritsch–Chappell (FC) closure (Fritsch and Chappell, 1980), the available
buoyant energy is dissipated during a specified convective time period (between 30 min and 1
hour).
Similarly, the Tiedtke (1989) scheme is a mass flux convection scheme, albeit it considers a
number of cloud types as well as cumulus downdrafts that can represent deep, mid-level and
shallow convection (Singh et al., 2011; Bhatla et al., 2016). The closure assumptions for the
deep and mid-level convection are maintained by large-scale moisture convergence, while the
shallow convection is sustained by the supply of moisture derived from surface evaporation.
**2.3 Numerical experiments and methodology**
Five experiments using the convection schemes of (1) Emanuel over land and Grell over ocean
(mix1), (2) Emanuel, (3) Grell, (4) Tiedtke and (5) Grell over land and Emanuel over ocean
(mix2) are conducted using RegCM4-CLM4.5 with 18 sigma levels at 50 Km horizontal
resolution for the period from November 2002 to September 2004. The two first months (i.e.
November and December 2002) was considered as spin-up time and not included in the
analysis. The analyses will focus on the rainy season from June to September (JJAS). As
quantitative measurements of model skills, we consider mean bias (MB) which is the difference



between the area-averaged value of the simulation and the observation, the spatial root mean
square difference (RMSD) and the spatial correlation called Pattern Correlation Coefficient
(PCC) and the distribution of Probability Density Function (PDF) of the temperature bias. The
RMSD, PCC and the PDF provide information at the grid-point level while the MB does so at
the regional level. A Taylor diagram (Taylor, 2001) is used to summarize assessments above
and to show the deviation of different model configurations results from observations.
As assumed in Gao et al., (2016), the temperature bias in JJAS present a normal mode type of
distribution. The PDF is expressed as:
$$\frac{1}{\sigma\sqrt{2\pi}}e\frac{(x-\mu)^2}{(2\sigma)^2}$$

Where $\mu$ is the mean and $\sigma$ the standard deviation of temperature bias.
The PDF is characterized by its bell shaped curve, the temperature biases distribute
symmetrically around the mean bias temperature value in decreasing numbers as one moves
away from the mean. The empirical rule states that for a normal distribution, nearly all of the
data will fall within three standard deviations of the mean. The empirical rule can be broken
down into three parts:
• 68% of grid points fall within the first standard deviation from the mean.
• 95% of grid points fall within two standard deviations from the mean.
• 99.7% of grid points fall within three standard deviations from the mean.
The rule is also called the 68-95-99.7 Rule or the Three Sigma Rule. Thus, they constitute
measurements of model performance and systematic model errors. These metrics are computed
for each of the sub-regions indicated in Figure 1.

**3 Results and discussion**
**3.1 Temperature**
The spatial distribution of averaged temperature during JJAS over 2003-2004 from CRU and
UDEL observations (resp. Fig. 2a, b) is compared to the temperature simulated by RegCM4
using the convection schemes: Mix1, Emanuel, Grell, Tiedtke and Mix2 (resp. Fig. 2c-g).
Figure 3 shows the associated mean model biases relatively to CRU for observation (UDEL;
Fig.3a) and the model simulations (Fig. 3b-f). Table 1 reports the PCC and the RMSD between
the simulated and observed temperature calculated for Guinea Coast, Central Sahel, West Sahel
and the entire West Africa domain. The CRU temperatures presents a zonal distribution in West
Africa with maximum (>34°C) in the Sahara and lowest temperatures (< 22°C) over the Guinea
Coast and over complex terrains such as the Jos plateau, Cameroon mountains and Guinean



highlands. The UDEL observation (Fig. 2b) shows similarity with CRU in terms of spatial
distribution with PCC larger than 0.98 over the entire West African domain (see Table 1).
However, UDEL depicts a sparse distribution with a mixture of warm and cold bias over the
Sahara and along of Nigeria/Cameroon border around ±2°C (see Fig. 3a). There is also a good
agreement between model simulated temperatures and CRU observation with the PCCs more
than 0.93 (Table 1) over West Africa. All model configurations well reproduce the general
features of the observed pattern including the meridional surface temperature gradient zone
between Guinea Coast and the Saharan desert. This temperature gradient is important for the
evolution of the African Easterly Jet (AEJ) (Cook 1999; Thorncroft and Blackburn, 1999). All
model configurations (Fig. 3b-d, f) exhibit a similar dominant cold biases except the Tiedtke
configuration (Fig. 3e) in the Sahara desert at the central part of Mauritania and Niger, and
along the Guinea Coast region. The greater cold bias with value up to -5°C occurs when using
Grell configuration while, simulation using Tiedtke configuration depicts a dominant warm bias
up to 4°C mainly located in Central Sahel around 12°N (Fig. 3e). One effect of the warm bias
shown in Tiedtke simulation is to shift the zone of meridional temperature gradient southward
relative to its observed position (Fig. 2f). However, it is difficult to determine the origin of
RCM temperature biases as they involve changes in surface-atmosphere interactions and as they
are function of many factors such as surface albedo, cloudiness, temperature advection and
surface water and energy fluxes (Tadross et al., 2006; Sylla et al., 2012).
For a quantitative evaluation of the performance of these sensitivity tests, the PDF statistical
tool was used. The PDF distributions of the temperature bias in JJAS is shown in Figure 4 for
Guinea Coast, Central Sahel, West Sahel and the entire West Africa domain. The PDF
distribution shows a general dominant cold bias (see Fig. 4a-d) in model simulations over most
of study domain, except with Tiedtke configuration in the Central Sahel region.
Over Guinea Coast region, Grell configuration presents a colder bias (reached -6°C) compared
to the other configurations. However, Emanuel simulation shows the lower RMSD about
1.29°C with a PCC larger than 0.77 (see Table 1). For Central Sahel region (Fig. 4b) a warmer
bias is found in Tiedtke simulation, while a colder bias is found in Grell and Mix2
configurations (see Fig. 4b). Emanuel configuration shows a lower value of RMSD about
0.67°C and a higher PCC larger than 0.95 compared to the other model simulated temperatures
(see table 1). In West Sahel a colder bias is found with Grell scheme (see Fig. 4c) while
Emmanuel and Tiedtke simulations show a mixture of cold and warm bias. Configuration of
RegCM with Emanuel presents a better performance with a lower RMSD and higher PCC
values compared to the other simulations in West Sahel. Over the entire West Africa domain



(see Fig.4d), Grell and Tiedtke present respectively a colder and warmer bias. Generally, with
respect to temperature simulations, a better performance of RegCM4 is obtained when using
Emmanuel scheme.

**3.2 Precipitation**

The spatial distribution of mean JJAS precipitation (2003–2004) over West Africa is shown in
Figure 5 for observations GPCP and TRMM (resp. Fig. 5 a-b) and for RegCM4 simulations
with the following convective schemes Mix1, Emmanuel, Grell, Tiedtke and Mix2 (resp. Fig.5
c-g). Sylla et al. (2013a) argued that over Africa, GPCP is more consistent with gauge based
observations, whilst Nikulin et al. (2012) found a significant dry bias over tropical Africa in
TRMM compared to GPCP. We therefore select, for precipitation, GPCP as our main
observational reference in this paper.

Figure 6 shows the corresponding precipitation mean biases relatively to GPCP for TRMM (Fig
6a) and for the different simulations configurations (Mix1, Emmanuel, Grell, Tiedtke and Mix2;
Fig 6b-f respectively). GPCP depicts a zonal band of rainfall decreasing from North to South.
Precipitation maxima are found in orographic regions of Guinea highlands, Jos Plateau, and
Cameroon Mountains. Differences between TRMM and GPCP observation products (Table 2)
can reach up to -5.26% at sub-regional levels, while over the entire West Africa it does not
exceed 0.82%. Although both observation products exhibit some differences (Fig.6a-c), their
patterns show a good agreement, with PCCs more than 0.96 over the entire West Africa domain
(Table 2). TRMM underestimates the rainfall intensity over Guinea Coast and Central Sahel
regions (respectively no more than -0.86% and -5.12%) and overestimates the rainfall intensity
over West Sahel and the entire West Africa domain reaching respectively 3.48% and 0.83%.
The spatial distribution of rainfall is well reproduced by all model configurations with PCCs
values within the range 0.61 and 0.89 over the entire West African domain. The dominant
feature in these simulations is the dry bias over West Africa domain (Fig. 6b-f), which is more
pronounced in the Tiedtke configuration (see Table 2). The warmer bias over Central Sahel in
Tiedtke configuration (Fig.3e) is consistent with the drier bias found in the same region (see
Table 2 and Fig.6e), as less rainfall would induce less evaporative cooling and increase the
insolation through decreased cloud cover. However, the Table 2 reveals that Mix1 and Emanuel
show a better performance with a lower mean biases and greater PCC compared to the others
model simulations over the entire West African domain and its sub-regions.



In order to understand the origins of the model rainfall biases, we analyzed the JJAS midlevel
(850–300 hPa) vertically integrated water vapor mixing ratio and the 650 hPa low-level wind
(African Easterly jet, AEJ) over West Africa averaged over the 2003–2004 period (Fig. 7). The
AEJ is the most prominent feature affecting the West African Monsoon through its role in
organizing convection and precipitation over the region (Cook 1999; Diedhiou et al., 1999;
Mohr and Thorncroft, 2006; Sylla et al., 2011). Areas with larger water vapor mixing ratio
corresponds to the areas of maximum precipitation in observations (see Fig. 5a-b). Around 9°N
the weaker easterly wind (AEJ) contributes to enhance the moisture convergence which results
in an increase of water vapor and precipitation (see Fig. 5a-b). All model configurations show
some quantitative differences compared to ERA-Interim in both the wind flux and the water
vapor mixing ratio.
The underestimation of vertically integrated water vapor mixing ratio is larger in Grell and
Mix2 simulations (Fig. 7 c, e) over the Guinea Coast and Atlantic Ocean compared to those of
Mix1, Emanuel and Tiedtke (Fig. 7 a, b, e). Mix1 and Emanuel configurations reproduce better
the spatial extent of the moisture convergence than the others model configurations (Fig.7b, c).
All model configurations simulate a stronger easterly wind flux (AEJ) than observed in
particular over the Guinea Coast and Atlantic Ocean inducing a negative impact on simulated
precipitations in the sub-regions (see Fig. 5c–g).    Another possible explanation of model
rainfall biases is further discussed in Brown and Sylla (2011) whereby a sensitivity study on
the domain size with RegCM3 over West Africa showed that RegCM3 simulates drier
conditions over a default domain (RegCM-D1) quite similar to our domain size used in this
study.
A Taylor diagram is used to give a combined synthetized view of the pattern correlation
coefficient and the JJAS standard deviation of precipitation from the different sensitivity studies
with respect to GPCP over Guinea Coast, Central Sahel, West Sahel and West Africa.  Model
standard deviations are normalized by the observed value from GPCP (indicated by REF, see
Fig.8). For the entire West Africa domain, the diagram shows Tiedtke and Emanuel outperform
the others configurations with values of standard deviation normalized much closer to 1.
However Emanuel configuration present a better spatial correlation reaching 0.8 as compared
to Tiedke configuration. Over Guinea Coast sub-region Grell and Emanuel present better values
of standard deviation normalized. However, in regarding the spatial correlation value about 0.7
Emanuel configuration is the best. For West and Central Sahel, Mix1 and Emanuel are closer
to observation. However, Emanuel outperforms Mix1 configuration with a good spatial
correlations scores between 0.7 and 0.8 respectively over Central and West Sahel sub-regions.



From the Taylor diagram, it can be inferred that Emanuel performs better regarding the standard
deviation normalized and the pattern correlation over the entire West African domain and its
sub-regions.
Based on previous experience and studies, Gao and al., (2016) noted that use of the Emanuel
convection scheme in RegCM3 and RegCM4 over China tends to simulate too much
precipitation when using BATS as the land surface scheme. They explained that it is mainly
due to the fact that the Emanuel scheme responds quite strongly to heating from the surface
land as compared to Grell and Tiedtke convection schemes, once convection is triggered. BATS
with only two soils levels depth maximizes this response; this is why Emmanuel is too wet
when using BATS as compared to Grell and Tiedtke. By contrast, CLM uses several soil layers
down to a depth of several meters; therefore, the upper soil temperatures respond less strongly
to the solar heating. Precipitation amount is much reduced when using CLM, which is good for
Emanuel but not good for Grell and Tiedtke (Gao and al., 2016) while the combination of BATS
with Grell and Tiedtke shows good performance (Gao et al., 2012; Ali et al., 2015).
In conclusion, although RegCM4-CLM4.5 shows some weaknesses, such as a dry bias over
most of Central Sahel and Guinea Coast region, its performance in replicating the spatial
distribution of rainfall appears in line with that documented in previous studies using the
previous version RegCM3 (Sylla et al. 2009; Abiodun et al.2012).

**3.3 Mean annual cycle**

In this section, we examine the effect of the convection scheme in the characterization of the
three distinct phases of the West African Monsoon: the onset, the high rain period and the
southward retreat of the monsoon rain band (Sultan et al., 2003). Such behavior is best
represented by a meridional cross-section (time-latitude Hovmoller diagram). This diagram
provides a robust framework to assess RCM's skills in simulating seasonal and intraseasonal
variations of the WAM, and thus the mechanisms of the region's rainfall (Hourdin et al., 2010).
Figure 9 shows the time-latitude diagrams of rainfall averaged over the region between 10°E
and 10°W for observations GPCP and TRMM (resp. Fig 9a-b) and for model simulations using
Mix1, Emmanuel, Grell, Tiedtke and Mix2 convection schemes (resp. Fig 9c-g). The averages
are taken for the period 2003–2004 and displayed throughout the year. This figure shows that
the three distinctively of monsoon phases are well represented by TRMM than GPCP (resp.
Fig.9a, b). TRMM observation shows a first rainy season from mid-March up to mid-June over
the Gulf of Guinea and Guinea Coast with a northward extension of the rain belt up to about
5°N (Fig.9b). The monsoon jump is characterized by a sudden cessation of precipitation



intensities (Sultan and Janicot, 2000, 2003) and occurs from mid-June to early July, when the
rain band core moves suddenly northward to about 10°N (Fig.9b). This indicates the beginning
of the rainy season over the Sahel with a peak reached in August between 9° and 12°N over
Central Sahel.  A gradual retreat of the monsoon starts in end of August and it is well shown by
GPCP (Fig.9a), with a decrease in intensity and a southward migration of the rain band. There
are both similarities and differences across the two observation datasets TRMM and GPCP.
Both datasets agree in area of rainfall maximum intensity around 4°N despite a more intense
peak of rainfall for TRMM compare to GPCP (resp. Fig.9a, b). The monsoon jump
characterized by a discontinuity sharp is not well defined in GPCP compared to TRMM. In
addition, GPCP shows wet conditions during the retreat phase in July to September compared
to TRMM (Fig.9a, b).
Mix1, Emanuel, and Grell model configurations (resp. Fig.9c-e) capture the three phases of the
seasonal evolution of the WAM, while Tiedke and Mix2 simulations fail to reproduce them in
particular the rainy season over Central Sahel. However Emanuel and Mix model
configurations (resp. Fig. 6c, d) overestimate rainfall amounts during the two rainy seasons over
Guinea Coast, mostly as a result of an overestimate of the precipitation over the orographic
regions of Guinea highlands, Jos Plateau, and Cameroon Mountains. Mix1 and Mix2
configurations are respectively wetter and drier compared to the others model configurations
(resp. Fig. 9c, g). Generally, the three monsoon phases are well shown by Grell simulation,
albeit it is drier compared to the others model simulations.
Another analysis of the annual cycle consists of considering the area-averaged (land-only grid
points) value of monthly rainfall and temperature over the Gulf of Guinea, the Central Sahel
and the entire West African domain (Figures 10 and 11). This allows better identification of
rainfall and temperature minima and peaks. Figure 10a-d shows respectively the annual cycle
of precipitation averaged over Guinea Coast, Central Sahel, West Sahel and the entire West
African domain. Over the Guinea Coast (Fig 10a), both GPCP and TRMM observations show
a primary maximum in June and a secondary one in September. The Mix1 and Tiedtke model
configurations simulate an early first peak in May while Emanuel, Grell and Mix2
configurations well capture the observed peak in June. We note that model configurations well
reproduce the timing of the mid-summer break and second rainfall peak in September but they
underestimate its magnitude, although Mix1 simulation result is higher and much closer to
observations compared to the others model simulations.
In both Central Sahel and West Sahel, observations (GPCP and TRMM) display a dry spring
(from January to March) and winter (from October to December) and a wet summer (from June





to September) with a well-defined peak occurring in August. Model configurations reproduce
both phase of the annual cycle and the observed rainfall peak in August except Emanuel
configuration which shifts it in September over West Sahel region. Model simulations
underestimate the peak intensity compare to observations. However Mix1 configuration rainfall
peak is much closer to observation for both Central Sahel and West Sahel regions (resp. Fig
10b, d) compared to the others model simulations. Over the entire West African domain, the
annual cycle (Fig 10c) is smoother with a notable shift of the peak in September in the different
model configurations. All the model configurations underestimate the rainfall peak and shift it
in October. However, Mix1 and Emanuel model simulations are much closer to observed annual
cycle of precipitation compared to the others. In resume Mix1 simulation compared to the others
better reproduce the observed annual cycle of precipitation over the sub-regions and the entire
West African domain.
The annual cycles of temperature for Central Sahel, West Sahel and the entire West African
domain of Mix1, Emmanuel, Grell, Tiedtke and Mix2 convection schemes are shown in Figure
11b-d. The observations (CRU and UDEL) indicate a cooler winter from December to February
and warmer pre and post-monsoon periods with relative minima occurring during August.
While over Guinea Coast, both winter and post monsoon are cooler and only the pre monsoon
phase is warmer (Fig. 11a). Models configurations present similar seasonal variation of the
mean monthly temperature at 2 m compared to observations, but do exhibit some differences.
Over Guinea Coast model simulations underestimate the magnitude of the temperature
compared to observations. However, Tiedtke configuration is higher and much closer to
observations compared to the others model simulations throughout the year (Fig.11a). Over
Central Sahel region, Grell and Tiedtke capture well the seasonal variation from November to
June in particular the first peak in August compared to the others models simulations. During
the summer (JJAS) Emanuel and Mix1 quite well reproduce the observed precipitation annual
cycle (Fig.11b). Therefore, model simulations underestimate the seasonal variation of
temperature over the entire West African domain. Although Tiedtke simulation overestimates
the mid-summer break period, it is much closer to observed annual cycle of temperature
throughout the year compared to the others model simulations. Over the West Sahel, model
simulations quite well reproduce the annual cycle of temperature except Grell and Mix2
configurations in particular during the summer (JJAS). In summary Tiedtke simulation better
reproduce the observed annual cycle of temperature throughout the year over the sub-regions
and the entire West African domain compared to the others model configurations.



The divergences in the RCMs annual cycles arise mostly from their different abilities to
simulate the main features responsible of triggering and maintaining the WAM precipitation
(Gbobaniyi E. et al., 2013). Among them, we have the monsoon flow, the African Easterly Jet
(AEJ), the Tropical Easterly Jet (TEJ) and the Africa Easterly Waves (AEWs) (Diedhiou et al.,
1999; Sylla et al., 2013b). .

**3.4 Wind profile**
The atmospheric circulations and their interactions with ITCZ play an important role in the
WAM system (Nicholson 2013). Thus, this section aims to analyze the impact of the choice of
convection scheme in the simulations of zonal winds features, including the near-surface
westerly component (the West African Monsoon, WAM), the African Easterly Jet (AEJ) and
the Tropical Easterly Jet (TEJ) in the mid and upper troposphere respectively. Figure 12 depicts
the vertical cross section of the JJAS mean of the zonal wind averaged between 10°W and 10°E
for ERA-Interim (Fig.12a) and model configurations in Mix1, Emmanuel, Grell, Tiedtke and
Mix2 convection schemes (resp. Fig.12 b-f). The reanalyse ERA-Interim (Fig. 12a) displays
the monsoon flow winds below 800 hPa at 2-18∘N with two cores merged over both Guinea
Coast (centered at 6°N) and Central Sahel (centered at 15°N) sub-regions, the AEJ in the mid-
levels centered at 12°N and the TEJ in the upper tropospheric levels at 200 hPa centered at 5°N
(Fig12 a). All model configurations well reproduce the zonal wind features despite some biases.
Model simulations Mix1, Emanuel, Grell and Tiedtke present a strong core of monsoon flow
compared to Era-Interim (reaching 6m/s). The stronger and weaker monsoon flows are found
with Mix1 and Mix2 configurations respectively compared to the others configurations.
However, model simulations well reproduce the limit of the surface westerly flow compared to
its position. Of particular interest is the core of the AEJ in the mid-tropospheric levels, which
is greatly weakened with Mx1 and Emanuel. While AEJ magnitude core is well defined in Grell
and Mix2 simulations at 12°N, but its spatial extent is somewhat reduced. This location of the
AEJ in Grell and Mix2 simulation is consistent with the location of the region of zonal
temperature gradient (see resp. Fig. 3e, g), as the AEJ is associated with the surface temperature
gradient (Cook 1999; Thorncroft and Blackburn 1999). While Tiedtke simulation shifts the
location of AEJ core at 8°N in agreement with the warm bias shown in Tiedtke configuration
(see Fig.4e). The TEJ at 200 hPa and 5°N is very similar in model simulations compared to the
ERA-Interim reanalysis. However, the core of the jet is weaker in Tiedtke configuration
compared to the others model simulations. An overall, Grell configuration outperforms
simulations of the main features of the zonal wind compared to the others model simulations.





**4 Summary and conclusion**

The latest released RegCM4 have been performed over West Africa for two years (2002-2003) to assess its performance using five convective parameterizations: (a) the Emanuel scheme, (b) Emanuel over land and Grell over Ocean (Mix1), (c) the Grell scheme, (d) the Tiedke scheme and (e) Grell over land and Emanuel over Ocean scheme (Mix2). The sensitivity of the model to different convection schemes were validated using observations. The main findings and conclusions can be summarized as follows:

(1) Compared with the previous version of RegCM, RegCM4-CLM also shows a general cold bias over West Africa. However in Central Sahel region, Tiedtke simulation presents a warm bias. This warms bias tends to displace the meridional temperature gradient southward relative to its observed position. An overall, with respect to temperature, better performance are obtained when using Emanuel scheme.

(2) With respect to the precipitation, the dominant feature in model simulations is a dry bias which is more pronounced when using Tiedtke convection scheme. Considering the good performance over the entire West Africa domain and its sub-regions in the temperature and precipitation simulations, we suggest Emanuel convection scheme when using RegCM4-CLM4.5 over West Africa.

(3) Simulations when using Mix1 and Emanuel schemes well reproduce the spatial extent of moisture convergence of the ERA-Interim reanalyses compared to the others convection schemes. However, in the mid-levels of the atmosphere, model simulations show an easterly wind flux (AEJ) stronger than observed in particular over the Guinea Coast and Ocean Atlantic below the latitude 4°N, creating an increased subsidence and has a negative effect on simulated precipitations there. This is a possible explanation of a dry bias over West Africa domain. However, the vertical features of the zonal wind, including the near-surface westerly component, the AEJ and the TEJ in the mid and upper troposphere are better simulated when using Grell convection scheme compared to the others model simulations

(4) The time evolution of simulation when using Grell convection scheme rainfall matches well with the observed evolution, including the timing of the discontinuous northward jump of the main rainfall band in late June, albeit it is drier compared to Mix1 and Emanuel convection scheme.

(5) Over Central Sahel and West Sahel, the mean annual cycle of precipitation and temperature, with the single peaked rainy season is especially well captured in terms of



505  timing despite the fact that all model simulations underestimated the magnitude.
506  However, simulations using Mix1 reproduce better the annual cycle of precipitation
507  compared to the others schemes.

508 (6) Over Guinea Coast, Mix1 and Tiedtke model simulations failed to reproduce the double
509  peaks rainy seasons, while Emanuel, Grell and Mix2 simulations well reproduce them
510  but underestimate their amplitude. The bimodal nature of rainfall associated with the
511  Guinea sub-region is not so well defined when averaging rainfall over the entire West
512  African domain. This emphasizes the importance of separating regions into
513  homogeneous precipitation sub-regions for evaluation analyses.

514 (7) The mean annual cycle of temperature, is well reproduce in simulation when using
515  Tiedtke convection scheme throughout the year over the sub-regions and the entire West
516  Africa domain compared to the others model simulations.

517

518 As more advanced package compared to the previously version of RegCM with BATS,
519 CLM4.5 can be considered as the primary land surface processes option in RegCM4.
520 Therein, the use of Emanuel scheme is recommended over the West African region. We
521 plan to use this configuration in long-term, multi-decadal simulations, to further evaluate
522 the model capability in reproducing the mean climatology, as well as the variability of
523 climate extremes over the region.

524

525 **Acknowledgements**
526 This work is dedicated to the memory of Prof Abdourahamane Konaré with whom we
527 started this assessment. The authors thank the Institute of Research for Development (IRD,
528 France) and Institute of Geosciences for Environment (IGE, University Grenoble Alpes) for
529 providing the facility (the Regional Climate Modelling Platform) to perform these
530 simulations and the IT support funded by IRD/PRPT contract at the University Felix
531 Houphouet Boigny (Abidjan, Côte d'Ivoire). The authors are grateful to all students,
532 technicians, engineers and researchers involved at ICTP (Abdus Salam International Centre
533 of Theoretical Physics; Trieste, Italy) on the development and the improvement of the
534 regional climate model RegCM.




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





|  | Guinea Coast | | Sahel Central | | West Sahel | | West Africa | |
|---|---|---|---|---|---|---|---|---|
|  | RMSD (°C) | PCC | RMSD (°C) | PCC | RMSD (°C) | PCC | RMSD (°C) | PCC |
| UDEL | 0.613 | 0.749 | 0.475 | 0.974 | 0.424 | 0.981 | 0.695 | 0.981 |
| Mix1 | 1.605 | 0.768 | 0.737 | 0.961 | 0.720 | 0.987 | 1.218 | 0.978 |
| Emanuel | 1.294 | 0.772 | 0.673 | 0.954 | 0.589 | 0.986 | 1.068 | 0.979 |
| Grell | 2.657 | 0.728 | 1.406 | 0.920 | 1.994 | 0.985 | 2.171 | 0.973 |
| Tiedtke | 1.534 | 0.758 | 1.360 | 0.938 | 0.717 | 0.982 | 1.355 | 0.938 |
| Mix2 | 1.993 | 0.781 | 1.682 | 0.884 | 1.568 | 0.978 | 1.715 | 0.964 |

**Table 1:** pattern correlation coefficient (PCC) and root mean square difference (RMSD) for JJAS 2m-temperature for model simulations and observation (UDEL) with respect to CRU over different sub-regions.

| | Guinea Coast | Sahel Central | West Sahel | West Africa | |
|---|---|---|---|---|---|
| | Mean Bias (%) | Mean Bias (%) | Mean Bias (%) | Mean Bias (%) | PCC |
| TRMM | -0.86 | -5.26 | 3.48 | 0.82 | 0.964 |
| Mix1 | -18.31 | -39.78 | -15.36 | -22.25 | 0.721 |
| Emanuel | -27.22 | -42.30 | -33.63 | -25.58 | 0.810 |
| Grell | -49.69 | -57.21 | -51.92 | -34.07 | 0.663 |
| Tiedtke | -48.71 | -77.14 | -56.67 | -65.08 | 0.613 |
| Mix2 | -54.43 | -50.55 | -55.42 | -47.66 | 0.897 |

**Table 2**: mean bias (MB) and the pattern correlation coefficient (PCC) for JJAS precipitation for model simulations and observation (TRMM) with respect to GPCP. The PCC is calculated only for the West African region.






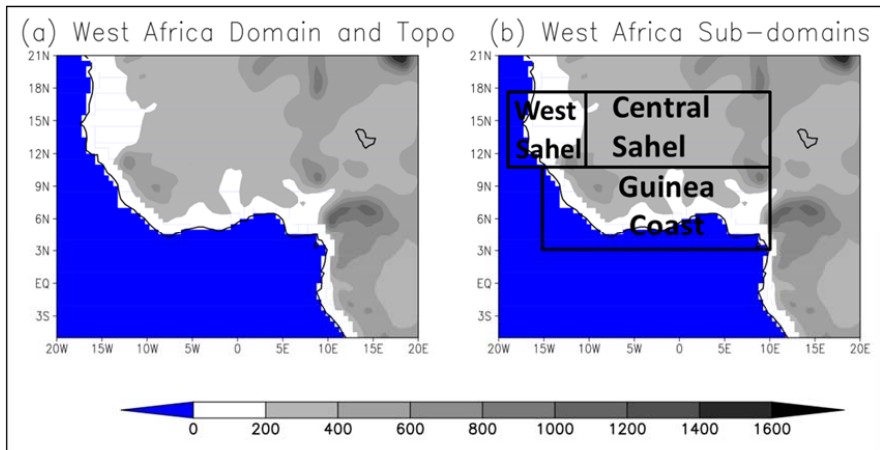



**Figure 1:** Domain, topography and sub-regions.

























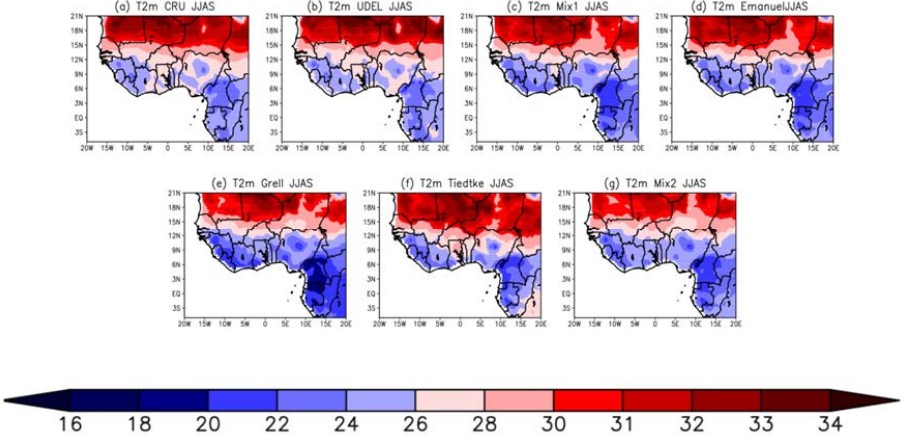



**Figure 2:** Averaged 2003–2004 JJAS 2m-temperature (in °C) from: (a) CRU, (b) UDEL,

(c) Mix1, (d) Emanuel, (e) Grell, (f) Tiedtke  and (g) Mix2.






















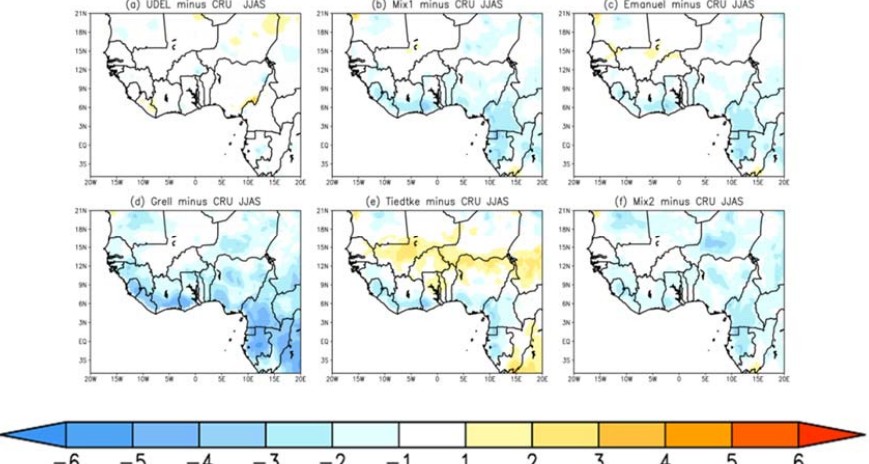



**Figure 3:** JJAS 2m-temperature bias (in °C) with respect to CRU from: (a) UDEL, (b) Mix1, (c) Emanuel, (d) Grell, (e) Tiedtke and (f) Mix2.











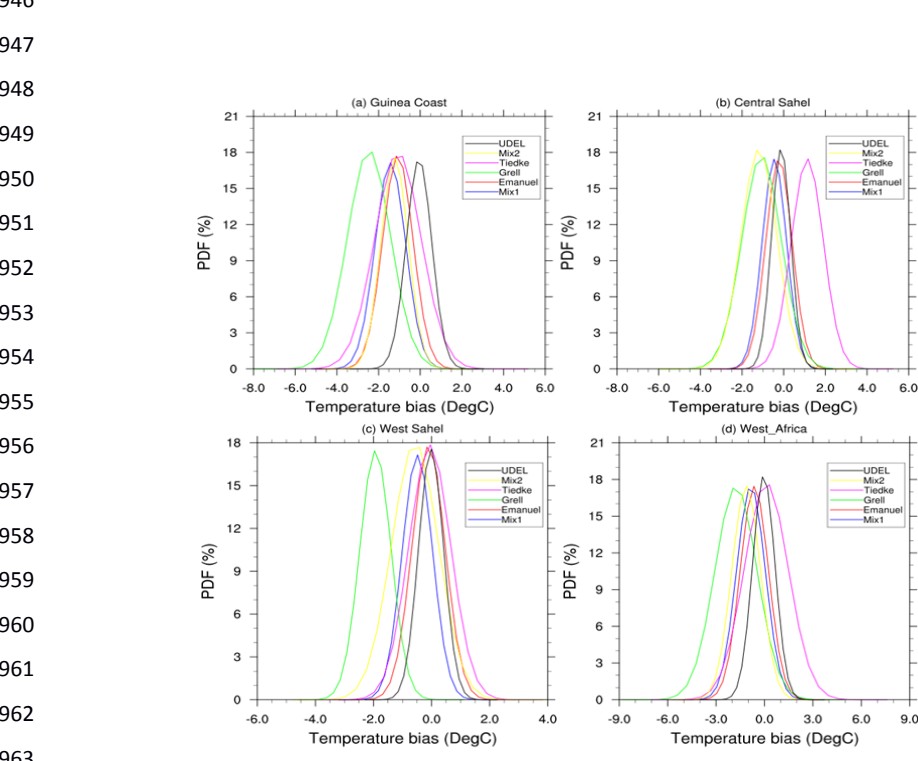

**Figure 4:** PDF distributions (%) of temperature bias in JJAS over Guinea, Central Sahel,
West Sahel and West Africa, derived from the model simulations using different convection
schemes (land only; units: °C).








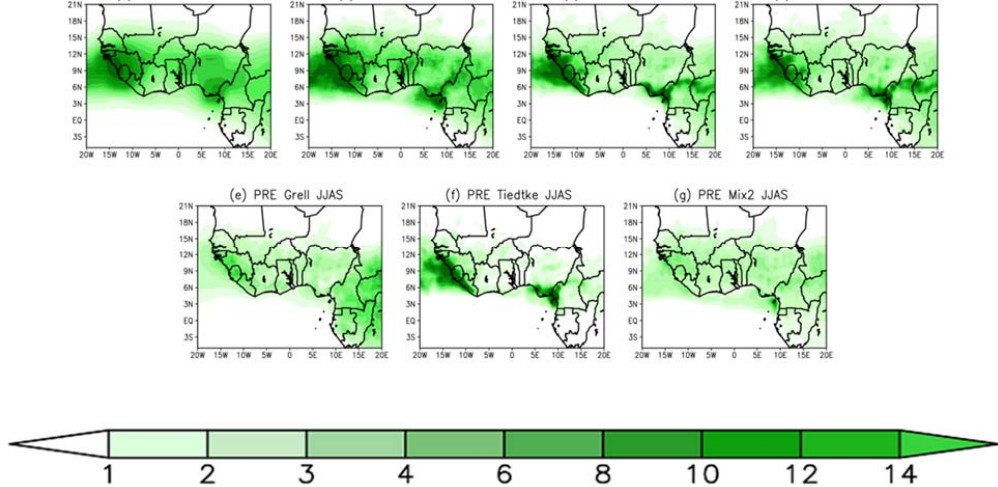



**Figure 5:** Averaged 2003–2004 JJAS precipitation (in mm/day) from: (a) GPCP, (b) TRMM,
(c) Mix1, (d) Emanuel, (e) Grell, (f) Tiedtke and (g) Mix2.



















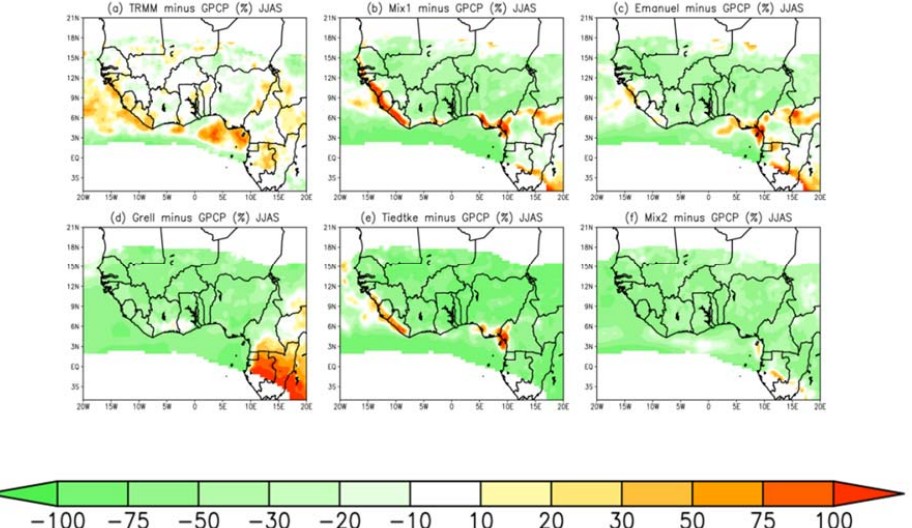



**Figure 6:** JJAS precipitation bias (in %) with respect to GPCP from : (a) TRMM, (b) Mix1, (c) Emanuel, (d) Grell, (e) Tiedtke and (f) Mix2.


























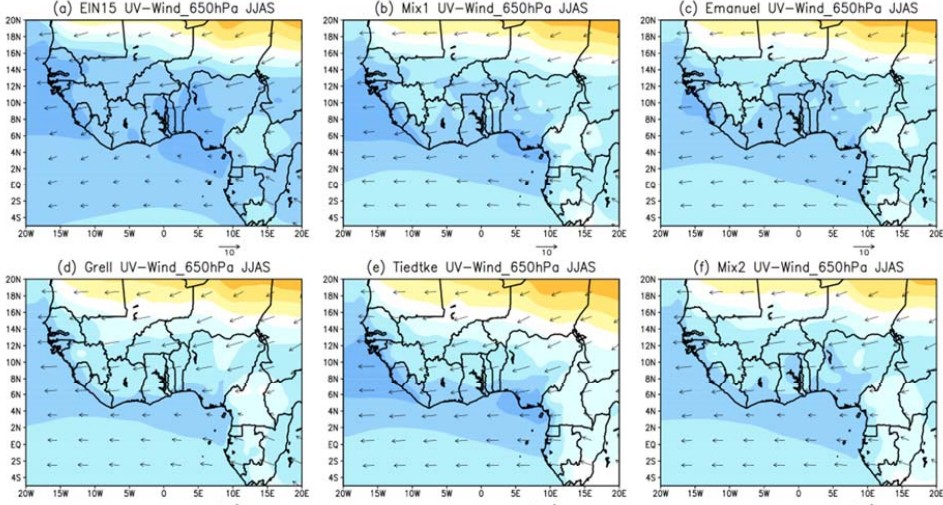


**Figure 7**: The (a) observed and (b–f) simulated vertically mean midlevel (850–300 hPa) integrated specific humidity (shaded) superimposed at zonal winds in JJAS at 650 hPa from (a) ERA-Interim, (b) Mix1 (c) Emanuel, (d) Grell (e) Tiedtke and (f) Mix2. Arrows are in m/s and specific humidity is expressed in $10^{-3}$ kg/kg.




















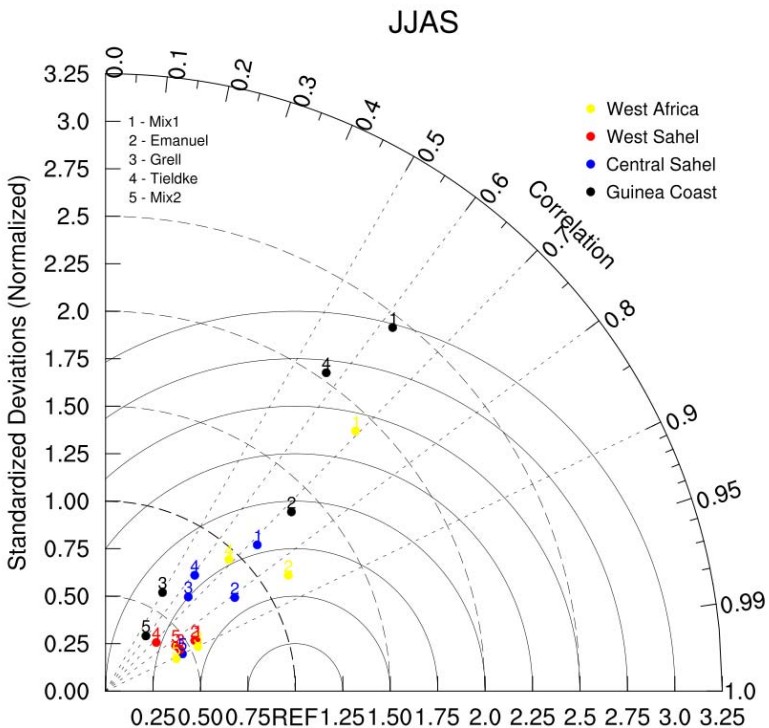


**Figure 8:** Taylor diagram showing the pattern correlation and the standard deviation (Normalized) of precipitation with respect to GPCP from: Mix1, Emanuel, Grell, Tiedtke and Mix2 over Guinea Coast, Central Sahel, West Sahel and West Africa.











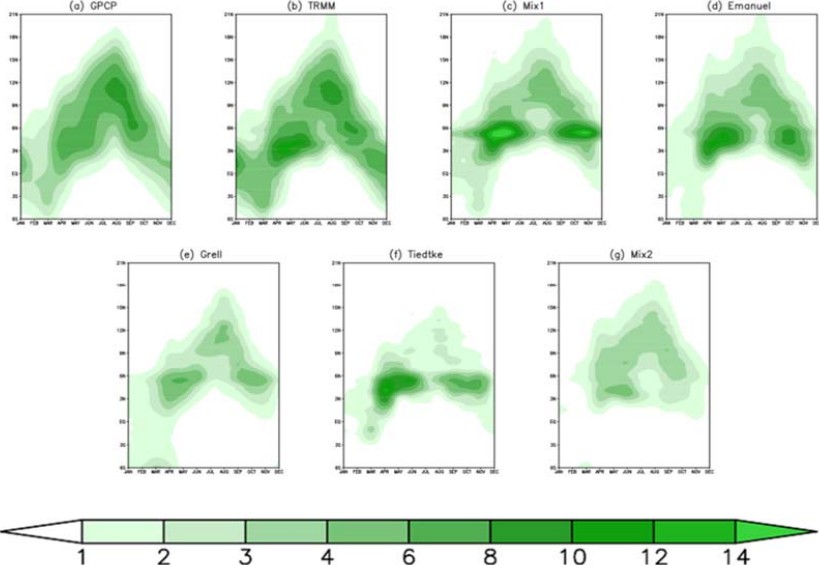



**Figure 9**: Hovmoller diagram of monthly precipitation (mm/day) averaged between 10°W
and 10°E and for the period 2003-2004 for (a) GPCP, (b) TRMM, (c) Mix1, (d) Emanuel,
(e) Grell, (f) Tiedtke and (g) Mix2.





















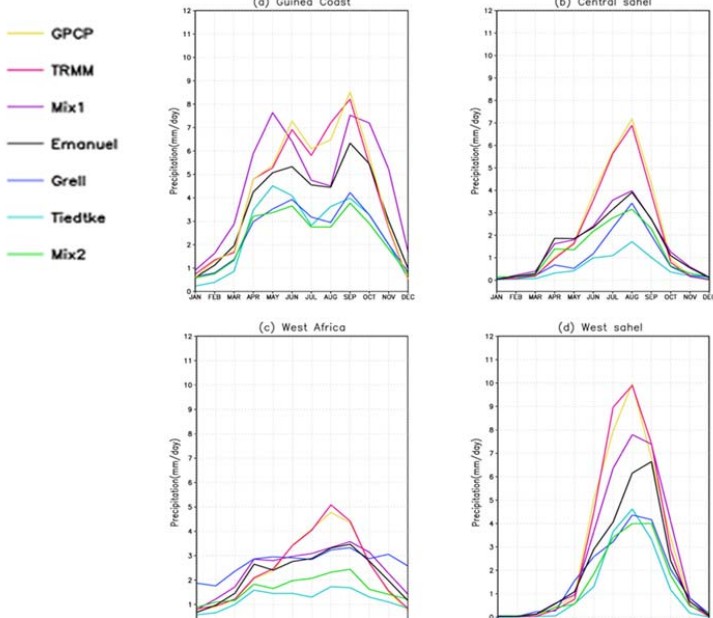




**Figure 10**: Annual cycle of monthly precipitation (mm.day$^{-1}$) averaged over, (a) the Guinea
Coast West and (b) Central Sahel, (c) West Africa and (d) West Sahel for the period 2003–

1086 2004.

















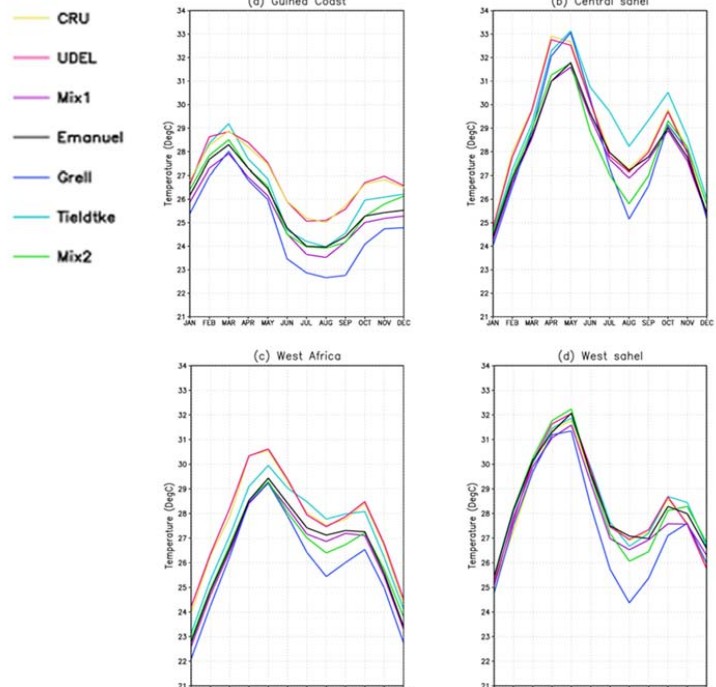



**Figure 11**: Annual cycle of 2m-Temperature (°C) averaged over, (a) the Guinea Coast, (b)
Central Sahel, (c) West Africa and (d) West Sahel for the period 2003–2004.

















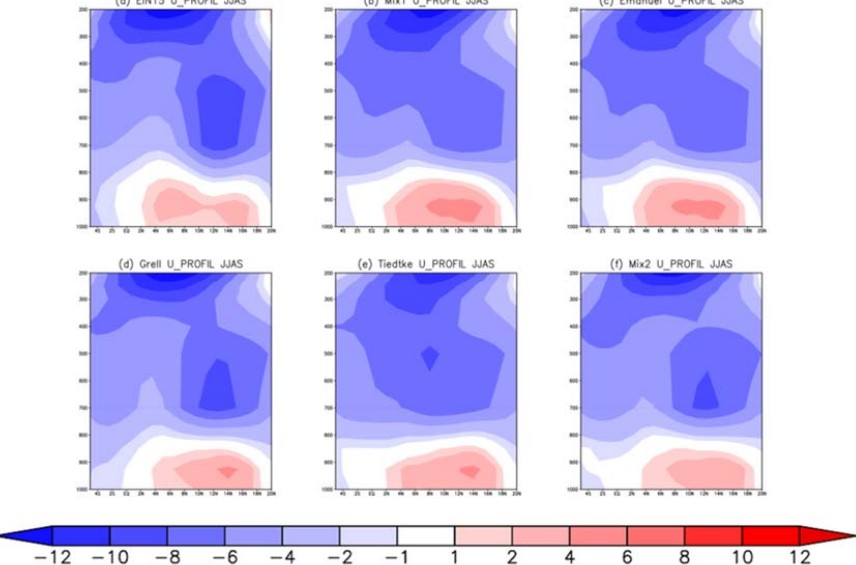



**Figure 12**: Vertical cross section of the JJAS mean zonal wind (in m/s) averaged between
10°W–10°E from: (a) ERA-Interim (b) Mix1, (c) Emanuel, (d) Grell, (e) Tiedtke and (f)
Mix2. The mean is calculated using the 2003–2004 period.
