# Peer review of "Sensitivity study of the Regional Climate Model RegCM4 to"

_Earth System Dynamics, 2018_

## Referee Comment (RC1) · Anonymous Referee #1 · 20 Jul 2018

This manuscript assessed the sensitivity of RCM RegCM4 with land surface component CLM4.5 to five convective schemes over West Africa from Nov. 2002 to Dec. 2004. With quantitative assessment sensitivity tests, results revealed better performance of the configuration with Emanuel convection scheme to simulate the air surface temperature and precipitation over West Africa by RegCM4-CLM4.5. Since CLM offers improvements in land-atmosphere exchange as you said, some mechanism analysis based on the water and energy fluxes of land-atmosphere interface should be added in the results part. There're some drawbacks which are detailed blow should be addressed before the paper can be published in ESD. Thus, a careful and rigorous revision is needed.

List of specific (major and minor) comments:

1. The abstract should present the main results rather than experiment setup in detail. Please rewrite the abstract.

2. Introduction section, P3, L73-77: You mentioned that RegCM-CLM led to weaker performance than RegCM-BATS over India, thus the performance of RegCM-CLM need to be assessed over West Africa? On the other hand, since CLM offers improvements in land-atmosphere exchange as you said, it might be better to give some mechanism analysis based on the water and energy fluxes of land-atmosphere interface in the results part. Additionally, many studies were found the model performs well using BATS over West Africa relative to CLM, could you present one result by RegCM-BATS to show how differences between them with same experiment setup?

3. P4, L115-117: many formats of citations are incorrect, such as Solmon et al., (2006) should be "Solmon et al. (2006)". Please revise the whole manuscript.

4. P4, L122: "Zeng and al., (1998)" should be "Zeng et al. (1998)"

5. P4, L130: "(Im et al., 2008);" should be "(Im et al., 2008;". L132: "Grell 1993" should be "Grell, 1993"

6. P5, L135-162: Some parts could be moved to 2.3 experiment setup.

7. P5, L158: Since RegCM uses Kilo-grid, it's not suitable to say all products are remapped onto RegCM4 grid (0.44ˆo\times 0.44ˆo).

8. P6, L196: What's is the model top?

9. P6, L197: In this study you adopt land surface component CLM4.5 which has more detailed description, thus need longer time to reach variable balance. Why do you just use two months run as spinup? And the computational cost is not too much.

10. P7, L210: "Where" should be "where" and assign number to the above equation and end with a comma.

[Figure]

11. P8, L258: "Grell configuration presents a colder bias (reached -6ˆoC)".

12. Figure 4: Could you explain why Tiedtke scheme show better performance in t2m than other convective schemes while it present larger winter bias in the central Sahel?

13. P9, L283-285: Which figure or reference support the conclusion "GPCP depicts a zonal band . . . Cameroon Mountains"

14. P9, L287: Fig.6a-c is Fig. 6a?

15. P9, L297-298: Please give more analysis or citation to support what you said.

16. P11, L365-367: From Figure 9a-b, you mentioned the three distinctively of monsoon phases are well represented by TRMM than GPCP. "well" is "better" and what's the criterion?

17. P14, L452: Format of superscript "o" seems wrong.

18. Revise all the captions of tables and figures with more details. And many texts in the figures are too small to see them.

19. Figure: Reduce the range of label bar to make those figures more comparable, such as range from 22 to 30. Same problem for Figure 3.

20. Since figure 3 show similar results with figure 2, consider removing one figure. Similar problem for Fig. 5 and 6.

21. Figure 7: No label bar. And consider using the differences rather than actual value.

---

## Referee Comment (RC2) · Anonymous Referee #2 · 30 Jul 2018

General Comments

This manuscript concerns the performance of the RegCM4 regional climate model in the simulation of present-day climate over West Africa, focusing on temperature and precipitation statistics. The most interesting issue addressed here is the investigation of whether using CLM4.5 as land surface scheme does add any value to the model's performances. Although the issue on sensitivity of convective schemes has been already investigated in the past (Sylla et al.,. . .: Djiotang et al., 2010; Komkoua et al., 2016; Adeniyi, 2014; and many others over the world), the subject is indeed interesting and deserves eventual publication on Earth System Dynamics.

[Figure]

However, before being acceptable for publication, there are some questions and clarifications that, in my opinion, have to be addressed.

1) Generally, any result that is used to support a statement about model performance (either positive or negative) should include a test of statistical significance (Fig. 3 and Fig. 6). Differences shown are significant?

2) Looking at precipitation extremes would be my first suggestion but the authors could look at other variables/statistics.

3) There is no justification for selecting 2003 and 2004 the analysis? Their motivations are not clear, Is there any particularities (dry, wet or normal) for those years?

4) Recents studies based were with RegCM simulation on a grid of 25km, in this manuscript there is no explanation on why they are running the simulation with a resolution of 50km (0.44°x0.44°).

5) My other concern is about the observation data used in this study, why they using the 2.5°x2.5° GPCP, instead of 1°x1° GPCP dataset? Why only GPCP, when other products like CHIRPS (0.01°x0.01°), ARC2 (0.1°x0.1°) are available and freely accessible.

6) For me, to give originality to this paper, authors should analyze diurnal cycle of rainfall.

7) Authors should convince the readers on the novelty of this research.
* * *

---

## Author Comment (AC1) · 4 Sep 2018

1- Comments from Referee1: the abstract should present the main results rather than experiment setup in detail. Please rewrite the abstract.

Author's response: Thank you for your comment. We rewrote the abstract in the manuscript.

Author's changes in manuscript: Please see at P1, L15-L34

2- Comments from Referee1: Introduction section, P3, L73-77: You mentioned that RegCM-CLM led to weaker performance than RegCM-BATS over India, thus the per-

formance of RegCM-CLM need to be assessed over West Africa? On the other hand, since CLM offers improvements in land-atmosphere exchange as you said, it might be better to give some mechanism analysis based on the water and energy fluxes of land-atmosphere interface in the results part. Additionally, many studies were found the model performs well using BATS over West Africa relative to CLM, could you present one result by RegCM-BATS to show how differences between them with same experiment setup?

Author's response: Thank you for your comment. The main objective of this study is to examine the performance of the last release of RegCM4-CLM4.5 over West Africa when using different convection schemes. We tried to identify the 'best' option with CLM4.5. It is not a performance assessment comparison between BATS and CLM4.5 in simulating the climate of West Africa. However, in the next (in a separate paper), we agree that we have to compare BATS and CLM4.5 over West Africa using RegCM4.4 because none of previous studies have attempted to investigate this over West Africa.

3- Comments from Referee1: many formats of citations are incorrect, such as Solmon et al., (2006) should be "Solmon et al. (2006)". Please revise the whole manuscript.

Author's response: Many thanks for your comment. Citations have been updated in this revised version.

Author's changes in manuscript: updated in several part of the text in this revised version

4- Comments from Referee1: P4, L122: "Zeng and al., (1998)" should be "Zeng et al. (1998)"

Author's response: Thank you. Citations have been checked and updated in this revised version.

Author's changes in manuscript: updated in several part of the text in this revised version

5- Comments from Referee1: P4, L130: "(Im et al., 2008);" should be "(Im et al., 2008;".
L132: "Grell 1993" should be "Grell, 1993".

Author's response: Many thanks. Correction done in this revised version

Author's changes in manuscript: Please see at P3, L134; P5, L139

6- Comments from Referee1: P5, L135-162: Some parts could be moved to 2.3 experiment setup.

Author's response: Yes, we agree. The paragraph has been moved as advised to 2.3

Author's changes in manuscript: Please see the manuscript at P7, L236 to P8, L264

7- Comments from Referee1: Since RegCM uses Kilo-grid, it's not suitable to say all products are remapped onto RegCM4 grid (0.44° * 0.44°).

Author's response: Thank you. We reformulate the sentence: "All products have been regridded to 0.44°×0.44° using a bilinear interpolation method to facilitate the comparison with RegCM4 simulations". This method is in line with those used by many authors in many papers such as Nikulin et al. (2012), Emiola et al. (2014), Diallo et al. (2014).

Author's changes in manuscript: Please see the manuscript at P7, L227

8- Comments from Referee1: P6, L196: What's the model top?

Author's response: The model top is 50 hPa. I mentioned it in the paper, please see at P7, L205.

9- Comments from Referee1: P6, L197: In this study you adopt land surface component CLM4.5 which has more detailed description, thus need longer time to reach variable balance. Why do you just use two months run as spin-up? And the computational cost is not too much.

Author's response: Thank you for your comment. Our choice of the spin up has been based on previous works. Suchul Kang and al. (2014) to justify the use of 7 days as

spin-up period to see the impact of two land-surface schemes on the characteristics of summer precipitation over East Asia from the RegCM4 simulations, examined the sensitivity of the length of spin-up by comparing the experiments with different spin-up period (7 days vs 1 month). They found that the difference of JJA mean precipitation between the two simulations seems to be random and trivial. In fact, Anthes and Al (1989) demonstrated that regional models attain the dynamical equilibrium between the lateral forcing and the internal physical dynamics of the model in about 2 –3 days. Also Gao and al. (2014) in comparison of convective parameterizations in RegCM4 experiments over China with CLM3.5 as the land surface mode, took one month spin-up. However in soil moisture simulation the spin-up must to be long, maybe more than one year as mentioned by Yang an al. (2011) in their work on effects of short spin-up periods on soil moisture simulation over New Zealand (J. Geophys. Res., 116, D24108, doi:10.1029/2011JD016121). In general 1 year simulation is adequate to evaluate model performance as mentioned by Gao and al., (2014) in his paper. That's why we used 2 years as simulation period and 2 months as spin-up in our work.

10- Comments from Referee1: P7, L210: "Where" should be "where" and assign number to the above equation and end with a comma.

Author's response: Yes we agree. We corrected it in the revised manuscript.

Author's changes in manuscript: Please see the manuscript at P6, L191–192

11- Comments from Referee1: P8, L258: "Grell configuration presents a colder bias (reached -6°C)".

Author's response: Many thanks for your comment. You are right. We are wrong. In fact, as shown in the Fig. 4a, -6 ° C bias temperature with Grell configuration is the temperature bias maximum at the grid level point, but its PDF distribution is very weak less than 1 %. We wanted to express that the maximum of the temperature bias distribution is centered on -2° C. This is in line with the RMSD (2.657 °C) calculated over Guinea Coast sub-region (see Table 1).
Author's changes in manuscript: The sentence has been modified in this revised version: "Over Guinea Coast region, Grell configuration presents a colder bias with the maximum of temperature bias distribution centered around -2°C (see Fig. 4a) compared to the other configurations". Please see at P9, L274-275..

12- Comments from Referee1: Figure 4: Could you explain why Tiedtke scheme show better performance in t2m than other convective schemes while it present larger winter bias in the central Sahel?

Author's response: Thank you for your comment. We are not sure to what sub-figure 4 you are referring. Tiedke is in pink and Emanuel is in orange. The last configuration performs better in Central Sahel as confirmed also in Table 1 when regarding RMSD and PCC values (resp. 0.673°C and 0.954) compared with Tiedke configuration (resp. 1.360°C and 0.938).

13- Comments from Referee1: P9, L283-285: Which figure or reference support the conclusion "GPCP depicts a zonal band . . . Cameroon Mountains"

Author's response: Thank you. "GPCP depicts a zonal band . . . . . .Cameroon Mountains" is supported by the Fig.5a

Author's changes in manuscript: We added the reference to the figure in this revised version. Please see at P9 L299-300

14- Comments from Referee1: P9, L287: Fig.6a-c is Fig. 6a?

Author's response: Thank you. Yes you are right. We corrected it in the manuscript.

Author's changes in manuscript: Please see at P9 L305.

15- Comments from Referee1: P9, L297-298: Please give more analysis or citation to support what you said.

Author's response: Thank you for your comment. The warmer bias over Central Sahel in Tiedtke configuration (Fig.3e) is consistent with the drier bias found in the same

region (see Table 2 and Fig.6e), as less rainfall would induce less evaporative cooling (decrease of latent heat flux) and therefore less favorable conditions for cloud cover (Feddema et al. 2005). The decrease of the cloud cover will lead to an increase of incident radiation inducing an increase of sensible heat flux and warmer surface temperatures. Moreover, a drier bias may be associated with a heating induced by the adiabatic subsidence to compensate effect of the increase of the surface albedo (Charney 1975).

Author's changes in manuscript: Please see at P10, L313-L320.

16- Comments from Referee1: P11, L365-367: From Figure 9a-b, you mentioned the three distinctively of monsoon phases are well represented by TRMM than GPCP. "well" is "better" and what's the criterion?

Author's response: Yes you are right, We rewrote the sentence. The criterion is that the cores of the different phases are well contrasted and reproduced in TRMM than in GPCP.

Author's changes in manuscript: Please see at P12, L389-L390

17- Comments from Referee1: P14, L452: Format of superscript "o" seems wrong.

Author's response: Thank you. We corrected it in this revised version

Author's changes in manuscript: Please see at P15, L476

18- Comments from Referee1: Revise all the captions of tables and figures with more details. And many texts in the figures are too small to see them. This range is the same used by many previous works.

Author's response: Thank you for the advice. We revised all the captions and text of tables and figures.

Author's changes in manuscript: Please see at several part if the text.

19- Comments from Referee1: Figure: Reduce the range of label bar to make those figures more comparable, such as range from 22 to 30. Same problem for Figure 3.

Author's response: The choice of this range is to make the figures comparable with previous works of Emiola Gbobaniyi et al. (2013), Ismaila Diallo et al (2014). However, in this revised version, the quality of the figure has been improved (see Fig3 and Fig.6).

20- Comments from Referee1: Since figure 3 show similar results with figure 2, consider removing one figure. Similar problem for Fig. 5 and 6. Author's response: Yes but the two figures are complementary. The spatial distribution of temperature and precipitation (resp. Fig 2 and Fig.5) shows the location of maximum and minimum values. While the spatial distribution of the biases (Fig.3 and Fig6), locates where the models deviate from the observation.

21- Comments from Referee1: Figure 7: No label bar. And consider using the differences rather than actual value.

Author's response: Yes you're right. Thank you, we have added the label bar in this revised version. We prefer to map the absolute values of the humidity to investigate how the spatial distribution of precipitation (mean meridional variation) is linked to the spatial variation of the humidity.

Author's changes in manuscript: Please see at P 35.

---

## Author Comment (AC2) · 4 Sep 2018

1. Comments from Referee2: Generally, any result that is used to support a statement about model performance (either positive or negative) should include a test of statistical significance (Fig. 3 and Fig. 6). Differences shown are significant?

Author's response: Thank you for your comment. You are right. We added a test of statically significance at 0.05 levels in this revised version. The test shows that the differences are statistically significant at 95%.

Author's changes in manuscript: Please see at P8, L247-L249; P9, L301-L303; P31
and P34 for figures.

2. Comments from Referee2: Looking at precipitation extremes would be my first suggestion but the authors could look at other variables/statistics.

Author's response: Thank you for your comment. That is right; We plan in a next study to investigate the sensitivity of temperature and precipitation extremes simulated by RegCM4-CLM4.5 to different convective schemes. In this paper, we are interested in the sensitivity of the mean climate.

Author's changes in manuscript: Please see at P17, L546-L550

3. Comments from Referee2: There is no justification for selecting 2003 and 2004 the analysis? Their motivations are not clear, Is there any particularities (dry, wet or normal) for those years?

Author's response: Thank you for your comment. The years 2003 and 2004 has been selected in this study because they corresponded respectively to dry and wet year in this region.

Author's changes in manuscript: Please see at P6 L179-L180

4. Comments from Referee2: Recent studies based were with RegCM simulation on a grid of 25km, in this manuscript there is no explanation on why they are running the simulation with a resolution of 50km (0.44°x0.44°).

Author's response: Thank you. The ICBC data in our possession were EIN15 (1.5° of resolution) and were appropriate for simulations at this standard resolution of 50Km to investigate the mean climate at regional scale. For the next study on the climate extremes, we will use EIN75 (0.75° of resolution) to perform simulations at 25 Km resolution.

5. Comments from Referee2: My other concern is about the observation data used in this study, why they using the 2.5x2.5 GPCP, instead of 1x1 GPCP dataset? Why only
GPCP, when other products like CHIRPS (0.01x0.01), ARC2 (0.1x0.1) are available and freely accessible.

Author's response: Thank you for your comment. That is right. In this revised version, we use GPCP 1°x1° dataset and TRMM dataset.

Author's changes in manuscript: Please see the Table 2 and the Fig. 9a

6. Comments from Referee2: For me, to give originality to this paper, authors should analyze diurnal cycle of rainfall.

Author's response: Thank you for your comment. We plan to investigate the diurnal in a second paper on the sensitivity of precipitation and temperature extremes with the same model at 0.25 Km resolution.

7. Comments from Referee2: Authors should convince the readers on the novelty of this research.

Author's response: Thank you for your comment. The originality of this study is in the fact that it is the first time RegCM4 with CLM as land surface scheme is assessed using a full set of convective schemes over Africa.

Author's changes in manuscript: Please see at P3 L91 -L103